# A Study on the Establishment of Physical Activity Environment for People with Disabilities in South Korea

**DOI:** 10.3390/healthcare10091638

**Published:** 2022-08-27

**Authors:** Ah-Ra Oh, Kyungjin Kim

**Affiliations:** 1Department of Exercise Rehabilitation & Welfare, Gachon University, 191 Hombakmoero, Yeonsu-gu, Incheon 21936, Korea; 2Department of Adapted Physical Activity, Korea Nazarene University, Cheonan-si 31172, Korea

**Keywords:** physical activity environment, people with disabilities, South Korea, relative importance, priority

## Abstract

The purpose of this study was to analyze the relative importance and priority of what factors should be reflected in the administration to efficiently consider the needs of people with disabilities for the physical activity (PA) environment in South Korea. To achieve the purpose of the study, 32 experts (e.g., faculty members, administrators) with more than 5 years of experience with PA for people with disabilities were asked to prioritize the factors that should be reflected in the PA environment. The questionnaire consisted of 4 factors in the upper-layer (H2), 8 factors in the middle-layer (H3), and 38 items in the low-layer (H4). The research instrument was composed of a pairwise comparison of decision elements to analyze the priority. For the analysis of the questionnaire data, the relative importance and priority were analyzed using Expert Choice 2000, a solution dedicated to priority analysis. The results are as follows. The relative importance of H2 was determined by programs, instructors, facilities, and information. In the relative importance among H3, the program type was determined as the highest factor in the program, and instructor expertise was determined as the highest factor in the instructor. The type of facility was determined as the highest factor in the facility, and the information provider was determined as the highest factor in the information. As a result of analyzing the priority of H4, it was found that the program within the sports facilities had the highest priority.

## 1. Introduction

Physical activity (PA) can be defined as any body movement generated by skeletal muscle that necessitates power expenditure [1]. According to the latest PA guidelines of the World Health Organization (WHO), adults around the world, including individuals with disabilities, benefit from health advantages and reduce health risks when they do more then 150 minutes of aerobic exercise per week with moderate intensity or vigorous intensity [1]. However, most adults around the world do not meet the PA guidelines of WHO [2]. On average, people with disabilities participate less in PA and are more sedentary than people without disabilities [3]. For instance, only about 50% of individuals with disabilities are active, compared to about 75% of people without disabilities in the United States [2]. In addition, only one-third of people with disabilities participate in sports, while two-thirds of individuals without disabilities participate in sports in the United States [4]. Only 12% of adults with physical disabilities participated in PA with moderate intensity or vigorous intensity [5].

According to data from the 2020 National Survey on the Current Status of Participation of Lifetime Sports for Persons with Disabilities, conducted by the Korea Ministry of Culture, Sports, and Tourism [6], the participation rate of PA for people with disabilities in South Korea was 24.9%, which is more than three times higher than 17.9% 10 years ago. Even though the participation rate of PA in 2021 has been lowered to 20.2% due to the Coronavirus Disease-19 (COVID-19) pandemic, this rate is expected to increase in the future [6]. The reason why PA is encouraged around the world is that engaging in PA immediately produces positive results for adults with disabilities [7]. It has been found that being provided opportunities to engage in PA can improve the quality of life for individuals with disabilities both physically and psychologically [8]. Moreover, PA participation among people with disabilities can prevent diseases and help to save the national health insurance finance [9].

Although the participation in PA of individuals with disabilities has many advantages, it is very low compared to the 60.8% participation rate of people without disabilities in South Korea [6]. Many people with disabilities participate in PA near or at home rather than at public PA-related facilities or social sports clubs for persons with disabilities [10]. However, as for the place of PA for people with disabilities in South Korea, the highest percentage was in a park or hiking trail near their home (61.5%), followed by their home (31.8%) [6]. Many people with disabilities participate in PA near or at home rather than at public PA-related facilities or social sports clubs for persons with disabilities, but there is little service support for individuals. The reason for this issue is that the analysis of the PA environment was insufficient and the needs of people with disabilities were not reflected [11].

In South Korea, studies are continuing to identify the needs of people with disabilities to facilitate their participation in PA or to investigate factors that make it difficult for them to actively participate in PA [11,12,13,14,15]. Summarizing the results of these studies, it is argued that, for the people with disabilities to participate more actively than previously in PA, their inner will to participate is significant, but rather than that, they demanded that the environment for participation in PA should be improved. Among the demands of persons with disabilities, the problems related to the expansion of public PA-related facilities and movement were found to be the biggest [6]. However, it cannot be concluded that there is not enough public PA-related facilities for people with disabilities to exercise at home and around the house. Participation in PA at home or around the house may be due to the characteristics of the type of disability, or it may be because the average age of the persons with disabilities has increased, or it may be because they simply enjoy taking a walk around the house. Thus, it is necessary to understand the cause from various angles and viewpoints from the perspective of individuals with disabilities, beyond the simple structure of expanding the public PA-related facilities for people with disabilities. In addition, creating an environment required by persons with disabilities will increase the satisfaction of people with disabilities and will continue to participate in PA.

Since the 1990s, the Australian Government has launched several policies to increase the participation of people with disabilities in recreation and sports [16]. Based on numerous research results, the Australian Government has made efforts to improve the environment and accessibility of open spaces, sports facilities, leisure, and swimming pools [17]. In the studies of the United States and the Netherlands, lack of motivation and experience of participating in PA or sports were individual barriers to participation in PA, such as environmental barriers, transportation and access to public PA-related facilities and cost were problems [18,19,20,21]. In addition, Moran, Taliaferro, and Pate [22] stated barriers to participation in PA for people with disabilities include lack of training knowledge and lack of programs for persons with disabilities. These studies include a long research period and the PA environment of persons with disabilities. Though, studies in South Korea that actively reflect the needs of people with disabilities and PA services have not yet been provided. Therefore, the purpose of this study was to determine the relative importance and priority for establishing the PA environment based on the opinions on the PA of persons with disabilities presented in a pilot study [23] and to introduce methods for successful access to the PA of people with disabilities in South Korea.

## 2. Materials and Methods

### 2.1. Participants

The purpose of this study was to identify the priority for establishment of the PA environment for persons with disabilities by reflecting the needs of people with disabilities. The participants of this study selected 32 disabled PA experts who understood the purpose of the study and agreed to participate in this study. The participants consisted of 12 faculty members who majored in PA with more than 5 years of research experience and 20 administrators who had worked at the Korea Paralympic Committee for more than 5 years. The reasons why the participants of this study were selected as faculty members and administrators of the Korean Paralympic Committee were that they were key experts in planning and implementing the administration of PA for people with disabilities in South Korea. The purpose of the study was explained to the participants of this research, and the PA environment for people with disabilities that should be developed first to reflect the needs of individuals with disabilities was investigated. The demographic characteristics of experts participating in this study are in Table 1.

### 2.2. Research Instrument

The study extracted the relative importance and priority using the priority analysis method (i.e., analytic hierarchy process (AHP)) to investigate which needs of the people with disabilities in the field of PA and experts’ thoughts should be reflected in the administration for PA environment for persons with disabilities. The questionnaire was constructed in the form of a pairwise comparison questionnaire for priority analysis [24]. The AHP is an analysis method that is often used in the process of collecting the opinions of experts. It is one of the decision-making methodologies that utilize the knowledge, experience, and intuition of respondents through binary comparison between factors forming a hierarchical structure of decision-making [25]. In other words, it is used as a scientific method for the decision-making method to reach final decision-making by dividing the entire decision-making process into stages and then analyzing and interpreting them step-by-step [26,27].

The research instrument of this study was based on the results of a focus group interview (e.g., 35 persons with disabilities in South Korea) conducted by the pilot study [23] on the needs of people with disabilities for PA in South Korea. A total of 38 key factors were extracted, and a questionnaire was prepared. The hierarchical structure of the questionnaire was shown in Figure 1. A group of experts (e.g., 3 faculties specializing in PA and 3 PA administrators for people with disabilities) checked the facial validity and composition of the questions for the questionnaire. As a result, there was hierarchy (H)1 for the establishment of the PA environment for persons with disabilities. The questionnaire consisted of 4 factors (e.g., program, instructor, facility, information: H2) in the upper-layer and 8 sup-factors (e.g., type of program, content of program, expertise of instructor, competency of instructor, type of facility, operation of facility, provider of information and content of information: H3) in the middle-layer. In the case of pairwise comparison, the response score of participants was ‘equivalent’ from 1 and ‘average’ to ‘very important’ from 2 to 7, so the factors of both poles were compared and selected.

### 2.3. Research Process

For the investigation of the study, the researchers first found two faculties based on PA for people with disabilities and two PA administrators in Korean Paralympic Committee. After that, the participants were secured by a snowball sampling method in which experts were introduced to other faculty members and PA administrators in Korean Paralympic Committee. A schedule was set with experts, and the purpose and contents of the study were explained to them. Since the pairwise comparison questionnaire is not a familiar questionnaire, the response method of the pairwise comparison was explained in detail. Participants in the study were asked to participate in the questionnaire, and the researchers collected it immediately after the questionnaire. In addition, the period of this study was 3 months from December 2021 to February 2022.

### 2.4. Data Analysis

The AHP was applied to analyze priorities for establishing the PA environment for people with disabilities. A hierarchical analysis is a method of calculating the relative importance between factors, and after identifying the hierarchical structure, priorities are selected through the analysis process [26,28]. The consistency index (CI) was verified to evaluate the logical contradiction of responses. The consistency is greater as the criterion of CI is closer to 0. In this study, consistency evaluation was conducted based on a value of 0.1 or less, which was generally accepted by previous studies [29,30]. As for the CI, when 0.1 or less appeared, it was confirmed that the consistency of the expert group for each question was high. The Expert Choice 2000 program was used to analyze priorities in this study.

## 3. Results

### 3.1. The Results of the Relative Importance of H2

In the relative importance of H2, the program showed the highest ranking as follows, followed by the instructor, facility, and information. The weight of the program was derived 38.9%, instructor (34.5%), facility (17.7%), and information (8.8%). The CI of the experts’ responses was 0.07, indicating that the respondents’ perceptions of the questions met consistency. The results of the relative importance of H2 for the establishment of the PA environment for people with disabilities are in Table 2.

#### 3.1.1. The Results of the Relative Importance of the Program in H3

As for the relative importance of the program in H3, the type of program was the highest factor, followed by the content of the program. As for the degree of importance perception of priority, the weight for the type of program was analyzed as 79.6%, and the content of the program (20.4%). The CI of the experts’ responses was 0.001, indicating that the respondents’ perceptions of the questions met consistency. The results of the relative importance of the program among H3 are in Table 3.

#### 3.1.2. The Results of the Relative Importance of the Instructor in H3

As for the relative importance of the instructor in H3, the expertise of the instructor was the highest factor, followed by the competency of the instructor. As for the degree of importance perception of priority, the weight for the expertise of the instructor was analyzed as 73.1%, and the competency of the instructor (26.9%). The CI of the experts’ responses was 0.001. The results of the relative importance of the instructor among H3 are in Table 4.

#### 3.1.3. The Results of the Relative Importance of the Facility in H3

As for the relative importance of the facility in H3, the type of facility was the highest factor, followed by the operation of the facility. As for the degree of importance perception of priority, the weight for the type of facility was analyzed as 76.1%, and the operation of the facility (23.9%). The CI of the experts’ responses was 0.001. The results of the relative importance of the facility among H3 are in Table 5.

#### 3.1.4. The Results of the Relative Importance of the Information in H3

As for the relative importance of the information in H3, the provider of information was the highest factor, followed by the content of information. As for the degree of importance perception of priority, the weight for the provider of information was analyzed as 74.1%, and the content of information (25.9%). The CI of the experts’ responses was 0.001. The results of the relative importance of the information among H3 are in Table 6.

### 3.2. The Results of the Relative Importance of H4 by H3

The results of the relative importance of low-layer (H4: e.g., programs within sports facility, sports or exercise clubs, programs of public outdoor facility, non-face-to-face programs, programs by type of disability, inclusive programs, rehabilitation exercise programs, programs by sport event, programs by age group, home training programs, instructor who has 2 or more certificates, instructor for persons with disabilities, instructor who has a disability, instructor for each sport event, instructor for all, understanding disability and human rights education, teaching method education by type of disability, safety education for PA, practical education for each sport event, public sports facilities for persons with disabilities, dedicated sports facilities for persons with disabilities, private sports facilities for persons with disabilities, sports facilities by sport events for persons with disabilities, public sports facilities, use of school sports facilities, neighborhood simple sports facilities, education on understanding of disability for all employees and users, promotion of facilities and programs, participation in sports program vouchers, accept and correct complaints, operation of health counseling office, city and province sports association for people with disabilities, government, community, sports facilities for people with disabilities in the local community, welfare center for people with disabilities, information about the location and program of the sports facilities, health information to motivate PA participation, sports club information about people with disabilities, information about the center for physical fitness and health measurement) by H3 were analyzed for the establishment of PA environment for persons with disabilities.

#### 3.2.1. The Results of the Relative Importance of H4 by the Program in H3

As shown in Table 7, the results of the relative importance of H4 by the program in H3 included the composition of H4 by H3, weight, ranking, and CI. As a result of the analysis, first, as the relative importance of H4 in the type of program in H3, the weight for programs within sports facilities was 40.7%, sports or exercise clubs for individuals with disabilities (25.3%), programs of public outdoor facilities (24.7%), and non-face-to-face (e.g., YouTube, etc.) programs (9.3%). The CI of responses of the experts in this study was 0.01.

As for the results of the relative importance of H4 by the content of the program in H3, the weight for programs by type of disability was 30.2%, inclusive (i.e., persons with and without disability) programs (16.8%), rehabilitation exercise programs (16.4%), program by sports events (16.2%), programs by age group (12.5%), and home training programs (8.0%). The CI of responses of the experts in this study was 0.01. The explanation of H4 by the program in H3 is in Table 8.

#### 3.2.2. The Results of the Relative Importance of H4 by the Instructor in H3

As shown in Table 9, the results of the relative importance of H4 by the instructor in H3 included the composition of H4 by H3, weight, ranking, and CI. As a result of the analysis of instructor, first, as the relative importance of H4 in the expertise of instructor in H3, the weight for an instructor who has 2 or more certificates related to PA was 32.7%, instructor for persons with disabilities (29.2%), an instructor who has a disability (16.5%), instructor for each sport event (12.3%), and instructor for all (9.3%). The CI of responses of the experts in this study was 0.02.

As for the results of the relative importance of H4 by the competency of instructor in H3, the weight for understanding disability and human rights education were 36.2%, teaching method education by type of disability (33.4%), safety education for PA (15.4%), and practical education for each sport event (15.1%). The CI of responses of the experts in this study was 0.02. The explanation of H4 by the instructor in H3 is in Table 10.

#### 3.2.3. The Results of the Relative Importance of H4 by the Facility in H3

As shown in Table 11, the results of the relative importance of H4 by the facility in H3 included the composition of H4 by H3, weight, ranking, and CI. As a result of the analysis, first, as the relative importance of H4 in the type of facility in H3, the weight for public sports facilities for persons with disabilities was 27.8%, dedicated sports facilities for persons with disabilities (22.3%), private sports facilities for persons with disabilities (14.8%), sports facilities by sports events for persons with disabilities (13.7%), public sports facilities (8.5%), use of school sports facilities (7.7%), and neighborhood simple sports facilities (5.3%). The CI of responses of the experts in this study was 0.01.

As for the results of the relative importance of H4 by the operation of facility in H3, the weight for education on understanding of disability for all employees and users was 26.4%, promotion of facilities and programs (26.1%), participation in sports program vouchers (19.9%), accept and correct complaints (15.4%), and operation of health counseling office (12.3%). The CI of responses of the experts in this study was 0.01. The explanation of H4 by the facility in H3 is in Table 12.

#### 3.2.4. The Results of the Relative Importance of the Information in H3

As shown in Table 13, the results of the relative importance of H4 by the information in H3 included the composition of H4 by H3, weight, ranking, and CI. As a result of the analysis of information, first, as the relative importance of H4 in the provider of information in H3, the weight for city and province sports association for people with disabilities was 26.1%, government (25.4%), community (19.7%), sports facilities for people with disabilities in the local community (15.4%), and welfare center for people with disabilities (13.4%). The CI of responses of the experts in this study was 0.01.

As for the results of the relative importance of H4 by the content of information in H3, the weight for information about the location and program of the sports facilities was 37.6%, health information to motivate PA participation (23.2%), sports club information about people with disabilities (22.0%), and information about the center for physical fitness and health measurement (17.1%). The CI of responses of the experts in this study was 0.01. The explanation of H4 by the information in H3 is in Table 14.

### 3.3. The Results of the Priority of H4

The results of the priority of H4 were analyzed for the establishment of the PA environment for individuals with disabilities. There was the priority of 1st to 40th of all H4. The priority of 1st to 40th of all H4 is in Table 15. The priority was as follows: programs within sports facilities (12.6%), instructor who has 2 or more certificates related to PA (8.2%), sports or exercise clubs for individuals with disabilities (7.9%), programs of public outdoor facilities (7.7%), instructor for persons with disabilities (7.4%), instructor who has a disability (4.2%), public sports facility for persons with disabilities (3.7%), education on understanding of disabilities and human rights (3.4%), instructor for each sport event (3.1%), teaching method education by type of disability (3.1%) and etc. The CI of responses of the experts in this study was 0.04.

## 4. Discussion

The aim of the study was to determine the relative importance and priority for the establishment of the PA environment for persons with disabilities by 32 experts with more than 5 years of experience with PA for people with disabilities. This study analyzed the relative importance and priority of establishing the PA environment based on the opinions on the PA of persons with disabilities offered in the pilot study. The relative importance and priority of establishing the PA environment for people with disabilities were analyzed by AHP. The discussion based on the results of the study was as follows; first, the results of the relative importance of H2 for the establishment of the PA environment for persons with disabilities was the rankings of program, instructor, facility, and information. The program was chosen as the highest factor in H2 to establish the PA environment for individuals with disabilities. However, there are no various PA programs for people with disabilities in South Korea. It may be helpful to motivate people with disabilities for participating in PA [23]. In both school and community settings, the PA programs for persons with disabilities are limited and require professional work [31]. For example, people with visual impairment cannot play regular baseball, but they can play it if they modify the game a little or modify the tool to put beads in the ball. Hence, modifying the program can be a successful key to participation in PA for people with disabilities [32]. After the program, it was found that the instructor was a key factor in establishing the PA environment. It was reported that learners’ behavior or attitudes were affected by instructors’ beliefs and values [33]. People with disabilities tend to rely on instructors due to various restrictions. This causes interference with the participation of persons with disabilities when the instructors negatively affect the participation of individuals with disabilities in PA [34]. The following factors included the facility and information. According to McGrath [8], it is important to provide playgrounds, leisure facilities, and swimming facilities to participate in sports and recreation for people with disabilities. In addition, the awareness of PA programs and sports facilities is essential for activating the PA of persons with disabilities [35]. The facilities should be built at the government level, and information can be obtained through individual efforts, so it is considered that the priority was lower than the factors of programs and instructors. 

Second, the results of the relative importance of the program in H3 were the type of program and content of the program. The results of rankings of the relative importance of H4 by the type of program in H3 were programs within sports facilities, sports or exercise clubs, programs of public outdoor facilities, and non-face-to-face programs. The results of rankings of the relative importance of H4 by the content of the program in H3 were programs by type of disability, inclusive programs, rehabilitation exercise programs, programs by sports event, programs by age group, and home training programs. The highest item of H4 by the type of program in H3 was the programs within a sports facility. There are various PA programs (e.g., aqua aerobics, badminton, basketball, football, golf, table tennis, squash, swimming, volleyball, and weight training) for individuals without disabilities in more than 1070 facilities and swimming facilities in South Korea. However, the PA programs (e.g., badminton, boccia, goalball, swimming, table tennis, weight training) for people with disabilities are only played at about 70 sports facilities in South Korea [36]. These programs limit the participation in the PA for persons with disabilities and cannot satisfy their needs. Furthermore, the highest item of H4 by the content of the program in H2 was the programs by type of disability because it is necessary to modify the PA program according to the characteristics of the type of disability as in the baseball example of people with visual impairment. According to Disability Rights in the United Kingdom, inclusive PA programs should be provided for persons with and without disabilities to participate in PA [23]. Additionally, when providing PA programs reflecting the needs of people with disabilities, voluntary participation of individuals with disabilities is expected to increase and satisfaction might increase [37].

Third, the results of the relative importance of the instructor in H3 were the expertise of the instructor and competency of the instructor. The results of rankings of the relative importance of H4 by the expertise of instructor in H3 were instructor who has 2 or more certificates, instructor for persons with disabilities, instructor who has a disability, instructor for each sport event, instructor for all. The results of rankings of the relative importance of H4 by the competency of instructor in H3 were understanding disability and human rights education, teaching method education by type of disability, safety education for PA, practical education for each sport event. The highest item of H4 by the expertise of instructor and competency of instructor in H3 were the instructor who has 2 or more certificates related to PA and understanding disability and human rights education. The professionalism (i.e., formal training, knowledge, resources) of PA instructors who know the characteristics of people with disabilities and PA is very important due to the activation of PA for individuals with disabilities [38]. A study [39] found the teaching of an instructor with expertise has an important influence on the effectiveness of PA, and the results of only 6% of the experience of PA with a professional instructor indicate what the future direction is. In addition, the professionalism of the instructor can increase the satisfaction of the participants by providing high-quality classes to people with disabilities [38]. Finally, PA instructors should have expertise and knowledge about people with disabilities to effectively teach them.

Fourth, the results of the relative importance of the facility in H3 were the type of facility and operation of the facility. The results of rankings of the relative importance of H4 by the type of facility in H3 were public sports facilities for persons with disabilities, dedicated sports facilities for persons with disabilities, private sports facilities for persons with disabilities, sports facilities by sports events for persons with disabilities, public sports facilities, use of school sports facilities, and neighborhood simple sports facilities. The results of rankings of the relative importance of H4 by the operation of the facility in H3 were education on the understanding of disability for all employees and users, promotion of facilities and programs, participation in sports program vouchers, accept and correct complaints, and operation of a health counseling office. The highest item of H4 by the type of facility and operation of the facility in H3 was the public sports facilities for persons with disabilities and education on the understanding of disability for all employees and users. As mentioned earlier, sports facilities for persons with disabilities in South Korea are insufficient compared to sports facilities for people without disabilities [36]. Several studies [40,41] reported that environmental factors such as accessibility and convenience of sports facilities are very significant for PA and securing of sports facilities is essential in relation to PA of people with disabilities. Conclusively, the lack of sports facilities for people with disabilities indicates that they are not even provided with opportunities to participate in PA, and the lack of understanding about the disability of the employees can even destroy their reliance to participate in PA.

Fifth, the results of the relative importance of the information in H3 were the provider of information and content of information. The results of rankings of the relative importance of H4 by the provider of information in H3 were city and province sports association for people with disabilities, government, community, sports facilities for people with disabilities in the local community, and welfare center for people with disabilities. The results of rankings of the relative importance of H4 by the content of information in H3 were information about the location and program of the sports facilities, health information to motivate PA participation, sports club information about people with disabilities, and information about the center for physical fitness and health measurement. The highest relative importance of H4 by the provider of information and content of information in H3 was the city and province sports association for people with disabilities and information about the location and program of the sports facilities. The studies of Pia [42] and Lyusyena [43] stated that the right to know about persons with disabilities is very important, and the PA of individuals with disabilities has a positive effect on reducing medical expenses and improving the quality of life. An efficient method for transmitting information is needed so that the city and province sports associations for people with disabilities oversee administrative work on PA for individuals with disabilities in South Korea and can know best about the information about the location and program of the sports facilities. Thus, the role of the city and province sports associations for people with disabilities is critical for effective information delivery to persons with disabilities.

Sixth, the results of the priority of H4 (i.e., the priority of 1st to 40th) were the programs within sports facilities, instructor who has 2 or more certificates related to PA, sports or exercise clubs for individuals with disabilities, programs of public outdoor facilities, instructor for persons with disabilities, instructor who has a disability, public sports facility for persons with disabilities, education on understanding of disabilities and human rights, instructor for each sport event, teaching method education by type of disability and so on. The highest priority of H4 for the establishment of the PA environment for persons with disabilities was the programs within sports facilities because there was a lack of sports facilities for people with disabilities, but also programs for individuals with disabilities in South Korea [36]. According to Brenda and Deborah [40], sports facilities are essential for individuals with disabilities to participate in sports and recreation, and the various programs provided by the sports facilities provide positive effects on the participation of persons with disabilities in PA. In addition, the next priority of H4 was the instructor who has 2 or more certificates related to PA. As stated earlier, instructors have an influence on the PA of people with disabilities; for instance, an instructor with expertise can effectively teach people with disabilities and provide high-quality teaching [38]. On the other hand, a small number of people with disabilities are provided with PA by professional instructors in South Korea [39]. The results of the relative importance and priority presented above showed similar results. Consequently, if the relative importance and priority are taken into consideration and the PA environment for persons with disabilities is established, successful PA can be provided for individuals with disabilities.

Therefore, these findings demonstrated that the programs, instructors, facilities, and information are all necessary for the successful establishment of the PA environment for individuals with disabilities. However, it will be possible to save time and to establish an effective PA environment for persons with disabilities based on the results of the relative importance and priority presented in this study.

## 5. Research Limitations

There are several limitations. First, this study is difficult to generalize because it was limited to specific participants (e.g., faculty members researching PA for people with disabilities, administrators in the Korean Paralympic Committee) in South Korea. Therefore, future studies should be conducted using data from various participants (e.g., persons with disabilities, administrators in the city and province sports association for people with disabilities). Second, the participants of this study were only Korean people with disabilities. Therefore, the establishment of the PA environment for people with disabilities according to cultural differences was not considered. Therefore, future research should be conducted in consideration of the PA environment for individuals with disabilities recommended by the world association for adapted PA or WHO. Third, this study analyzed the priority that should be reflected in the administration for the establishment of the PA environment for persons with disabilities, but there are no specific administrative plans presented in this study. Therefore, there is a need for a study that specifically suggests administrative plans that reflect the needs of individuals with disabilities and the opinions of experts.

## 6. Conclusions

The purpose of this study was to analyze the priority of what factors should be reflected in the administration to efficiently consider the needs of persons with disabilities for the PA environment in South Korea. To achieve the purpose of the study, 32 PA experts were asked about the factors that should be reflected in the PA environment first through the questionnaire in the pairwise comparison format. The conclusions shown through the hierarchical analysis are as follows; first, the relative importance between H2 was determined by program, instructor, facility, and information. Second, in the relative importance among H3, the type of program was determined as the highest factor in the program, and the expertise of the instructor was determined as the highest factor in the instructor. The type of facility was determined as the highest factor in the facility, and the provider of information was determined as the highest factor in the information. Third, the results of analyzing the priority of H4, it was decided as the program within a sports facility had the highest priority.

## Figures and Tables

**Figure 1 healthcare-10-01638-f001:**
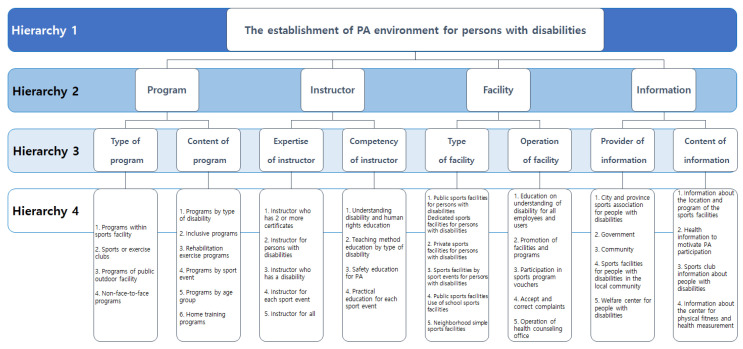
The hierarchical structure of the questionnaire.

**Table 1 healthcare-10-01638-t001:** Demographic characteristics of participants.

No.	Classification	Gender	Years of Experience	Age
1	Faculty	Female	11	34
2	Faculty	Male	10	41
3	Faculty	Male	14	54
4	Faculty	Male	5	32
5	Faculty	Male	5	39
6	Faculty	Female	5	33
7	Faculty	Male	17	51
8	Faculty	Male	8	48
9	Faculty	Male	8	39
10	Faculty	Male	5	40
11	Faculty	Male	6	32
12	Faculty	Male	5	46
13	Administrator	Male	13	48
14	Administrator	Male	21	40
15	Administrator	Female	11	45
16	Administrator	Male	14	39
17	Administrator	Male	21	48
18	Administrator	Male	10	42
19	Administrator	Male	6	42
20	Administrator	Male	11	40
21	Administrator	Male	14	42
22	Administrator	Male	12	41
23	Administrator	Female	5	29
24	Administrator	Female	5	25
25	Administrator	Male	7	35
26	Administrator	Female	8	36
27	Administrator	Female	13	42
28	Administrator	Male	10	41
29	Administrator	Female	5	29
30	Administrator	Male	5	30
31	Administrator	Male	24	49
32	Administrator	Female	9	36

**Table 2 healthcare-10-01638-t002:** Results of the relative importance of H2 for the establishment of the PA environment for persons with disabilities.

Classification	Factor	CI	Order
The establishment of PA environment for persons with disabilities	Program	0.389	1
	Instructor	0.345	2
	Facility	0.177	3
	Information	0.088	4

Note: CI = 0.07.

**Table 3 healthcare-10-01638-t003:** Results of the relative importance of the program in H3.

Classification	Factor	CI	Order
Program	Type of program	0.796	1
	Content of program	0.204	2

Note: CI = 0.001.

**Table 4 healthcare-10-01638-t004:** Results of the relative importance of the instructor in H3.

Classification	Factor	CI	Order
Instructor	Expertise of instructor	0.731	1
	Competency of instructor	0.269	2

Note: CI = 0.001.

**Table 5 healthcare-10-01638-t005:** Results of the relative importance of the facility in H3.

Classification	Factor	CI	Order
Facility	Type of facility	0.761	1
	Operation of facility	0.239	2

Note: CI = 0.001.

**Table 6 healthcare-10-01638-t006:** Results of the relative importance of the information in H3.

Classification	Factor	CI	Order
Information	Provider of information	0.741	1
	Content of information	0.259	2

Note: CI = 0.001.

**Table 7 healthcare-10-01638-t007:** Results of the relative importance of H4 by the program in H3.

Highest Factor	Sub-Factor	Sub-Item	Weight	Order	CI
Program	Type of program	Programs within sports facility	0.407	1	0.01
		Sports or exercise clubs	0.253	2	
		Programs of public outdoor facility	0.247	3	
		Non-face-to-face programs	0.093	4	
	Content of program	Programs by type of disability	0.302	1	0.01
		Inclusive programs	0.168	2	
		Rehabilitation exercise programs	0.164	3	
		Programs by sport event	0.162	4	
		Programs by age group	0.125	5	
		Home training programs	0.080	6	

Note: CI = 0.001.

**Table 8 healthcare-10-01638-t008:** Explanation of H4 by the program in H3.

Sub-Factor	Sub-Item	Explanation of Item
Type of program	Programs within sports facility	PA programs for people with disabilities in public and private sports facilities
	Sports or exercise clubs	PA programs in sports or exercise clubs for people with disabilities
	Programs of public outdoor facility	PA programs using public sports facilities in local parks
	Non-face-to-face programs	Non-face-to-face PA programs that can be performed at home
Content of program	Programs by type of disability	PA programs by type of disability
	Inclusive programs	PA programs for people with and without disabilities
	Rehabilitation exercise programs	Rehabilitation exercise programs that serve as a bridge to PA
	Programs by sport event	PA programs by each sport event
	Programs by age group	PA programs by age group
	Home training programs	PA programs that can be performed at home alone or with your family member

Note: CI = 0.001.

**Table 9 healthcare-10-01638-t009:** Results of the relative importance of H4 by the instructor in H3.

Highest Factor	Sub-Factor	Sub-Item	Weight	Order	CI
Instructor	Expertise of instructor	Instructor who has 2 or more certificates	0.327	1	0.02
		Instructor for persons with disabilities	0.292	2	
		Instructor who has a disability	0.165	3	
		Instructor for each sport event	0.123	4	
		Instructor for all	0.093	5	
	Competency of instructor	Understanding disability and human rights education	0.362	1	0.02
		Teaching method education by type of disability	0.334	2	
		Safety education for PA	0.154	3	
		Practical education for each sport event	0.151	4	

Note: CI = 0.001.

**Table 10 healthcare-10-01638-t010:** Explanation of H4 by the instructor in H3.

Sub-Factor	Sub-Item	Explanation of Item
Expertise of instructor	Instructor who has 2 or more certificates	An instructor who has 2 or more certificates related to PA for all, people with disabilities, or athletes
	Instructor for persons with disabilities	An instructor who has a PA certificate for persons with disabilities
	Instructor who has a disability	An instructor who has a disability
	Instructor for each sport event	An instructor for each sport event
	Instructor for all	A PA instructor for all
Competency of instructor	Understanding disability and human rights education	Understanding the types of disability and reduce derogatory comments and discrimination through education on the human rights of persons with disabilities
	Teaching method education by type of disability	Teaching instructional methods suitable for the type of disability and teach communication methods (sign language, etc.)
	Safety education for PA	Improving the ability to cope with accidents that may occur during PA (ex. emergency rescue, etc.)
	Practical education for each sport event	Upgrading skills and knowledge about each sport event

Note: CI = 0.001.

**Table 11 healthcare-10-01638-t011:** Results of the relative importance of H4 by the facility in H3.

Highest Factor	Sub-Factor	Sub-Item	Weight	Order	CI
Facility	Type of facility	Public sports facilities for persons with disabilities	0.278	1	0.01
		Dedicated sports facilities for persons with disabilities	0.223	2	
		Private sports facilities for persons with disabilities	0.148	3	
		Sports facilities by sport events for persons with disabilities	0.137	4	
		Public sports facilities	0.085	5	
		Use of school sports facilities	0.077	6	
		Neighborhood simple sports facilities	0.053	7	
	Operation of facility	Education on understanding of disability for all employees and users	0.264	1	0.01
		Promotion of facilities and programs	0.261	2	
		Participation in sports program vouchers	0.199	3	
		Accept and correct complaints	0.154	4	
		Operation of health counseling office	0.123	5	

Note: CI = 0.001.

**Table 12 healthcare-10-01638-t012:** Explanation of H4 by the facility in H3.

Sub-Factor	Sub-Item	Explanation of Item
Type of facility	Public sports facilities for persons with disabilities	A sports facility that people with disabilities prefer to use but can use together with people without disabilities
	Dedicated sports facilities for persons with disabilities	Sports facilities mainly used by people with disabilities
	Private sports facilities for persons with disabilities	A private sports facility that installs convenient facilities for people with disabilities and recruit members with disabilities
	Sports facilities by sport events for persons with disabilities	Sports facilities for people with disabilities specializing in sports
	Public sports facilities	Sports facilities created for residents
	Use of school sports facilities	School sports facilities opened as sports facilities for people with disabilities
	Neighborhood simple sports facilities	Sports equipment and facilities simply installed in community parks
Operation of facility	Education on understanding of disability for all employees and users	It is necessary to make it compulsory for employees and users to understand disability education so that users with disabilities can use it without discrimination
	Promotion of facilities and programs	Promotion of facility and program recruitment through text messages, SNS, website, etc. should be actively promoted
	Participation in sports program vouchers	It is necessary to prepare an environment in which sports program vouchers can be used.
	Accept and correct complaints	Efforts should be made to actively respond to and accept complaints when they are filed
	Operation of health counseling office	There is a need for a counseling office that can conduct and manage consultations on the health status of people with disabilities and how to maintain and improve their health in the future

Note: CI = 0.001.

**Table 13 healthcare-10-01638-t013:** Results of the relative importance of H4 by the information in H3.

Highest Factor	Sub-Factor	Sub-Item	Weight	Order	CI
Information	Provider of information	City and province sports association for people with disabilities	0.261	1	0.01
		Government	0.254	2	
		Community	0.197	3	
		Sports facilities for people with disabilities in the local community	0.154	4	
		Welfare center for people with disabilities	0.134	5	
	Content of information	Information about the location and program of the sports facilities	0.376	1	0.01
		Health information to motivate PA participation	0.232	2	
		Sports club information about people with disabilities	0.220	3	
		Information about the center for physical fitness and health measurement	0.171	4	

Note: CI = 0.001.

**Table 14 healthcare-10-01638-t014:** Explanation of H4 by the information in H3.

Sub-Factor	Sub-Item	Explanation of Item
Provider of information	City and province sports association for people with disabilities	At the level of city and province sports associations for people with disabilities, health information should be provided to individuals with disabilities in the local community by text message or mail
	Government	The government should provide health information for people with disabilities through the mass media
	Community	Health information should be provided by text message or mail at the city hall or community center in the local community
	Sports facilities for people with disabilities in the local community	Health information should be provided to users with disabilities and their families in sports facilities
	Welfare centers for people with disabilities	Welfare centers should provide health information to users with disabilities and their families
Content of information	Information about the location and program of the sports facilities	Information on where and what programs are available for people with disabilities in the community
	Health information to motivate PA participation	Motivation of PA by providing information on PA and eating habits
	Sports club information about people with disabilities	Location and related information about sports clubs for people with disabilities and PA
	Information about the center for physical fitness and health measurement	Information on physical fitness certification center for people with disabilities

Note: CI = 0.001.

**Table 15 healthcare-10-01638-t015:** Results of the priority of H4.

Item	Weight	Order	Item	Weight	Order
Programs within sports facilities	0.126	1	Programs by sport event	0.013	21
Instructor who has 2 or more certificates related to PA	0.082	2	Inclusive programs	0.013	22
Sports or exercise clubs for individuals with disabilities	0.079	3	Rehabilitation exercise programs	0.013	23
Programs of public outdoor facilities	0.077	4	Community	0.013	24
Instructor for persons with disabilities	0.074	5	General public sports facilities	0.011	25
Instructor who has a disability	0.042	6	Promotion of facilities and programs	0.011	26
Public sports facility for persons with disabilities	0.037	7	Education on understanding of disability for all employees and users	0.011	27
Education on understanding of disabilities and human rights	0.034	8	Programs by age group	0.010	28
Instructor for each sport event	0.031	9	Use of school sports facilities	0.010	29
Teaching method education by type of disability	0.031	10	Sports facilities for people with disabilities in the local community	0.010	30
Public sports facility for persons with disabilities	0.030	11	Welfare centers for people with disabilities	0.009	31
Non-face-to-face programs	0.029	12	Information about the location and program of the sports facilities	0.009	32
Programs by type of disability	0.024	13	Participation in sports program vouchers	0.008	33
Instructor for all	0.024	14	Neighborhood simple sports facilities	0.007	34
Private sports facilities for persons with disabilities	0.020	15	Accept and correct complaints	0.007	35
Sports facilities by sport events for persons with disabilities	0.018	16	Home training programs	0.006	36
Government	0.017	17	Operation of health counseling office	0.005	37
City and province sports association for people with disabilities	0.017	18	Health information to motivate PA participation	0.005	38
Practical education for each sport event	0.014	19	Sports club information about people with disabilities	0.005	39
Safety education for PA	0.014	20	Information about the center for physical fitness and health measurement	0.004	40

Note: CI = 0.04.

## Data Availability

The original contributions presented in the study are included in the article, further inquiries can be directed to the corresponding author.

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
