# Peer review of "A Study on the Establishment of Physical Activity Environment for People with Disabilities in South Korea"

_healthcare, 2022, doi:10.3390/healthcare10091638_

Round 1

Reviewer 1 Report

Congratulations

Very well prepared article of high scientific technical competence, use of the statistic very well done.

It would be of great importance to review the writing on English.

Att.

Author Response

Dear Reviewers,
We have revised a manuscript entitled “A Study on the Establishment of Physical Activity Environment for People with Disabilities in South Korea” for consideration by Healthcare. We confirm that this work is original and has not been published elsewhere nor is it currently under consideration for publication elsewhere.
For our revision, we feel that we have adequately addressed all of the comments from the reviewers. Please see our comments and edits for each of the reviewer’s comments in attached file. In this same correspondence, we have also uploaded two versions of our manuscript: a clean copy and a copy highlighting the edits and additions from reviewers. In this latter copy, all edits as suggested by the second reviewer are highlighted in blue and all edits as suggested by the third reviewer are highlighted in green. Any common edits (e.g., editing of English in the manuscript) from the reviewers are in yellow font.
Thank you for your consideration of this manuscript. Please address all correspondence concerning this manuscript to me at [email protected].

Please see after 20 pages.

Reviewer 2 Report

Overall, this study has good quality statements. Further, the subject is relevant, the sample size is considerable and an effort was made in terms of literature review and identification of research limitations. If only a few parts were modified, it would be a better study. Questions or corrections are as follows:

1. Writing needs some improvement and some minor spell check required.

2. Specify the duration of the study in the study procedure.

3. No mention of the study subject's intention to select the sample. Please explain why you selected the experts as the physical education administration and professor group for people with disabilities.

4. Add a sequence number to the category in <table 1>.

Author Response

(The authors gave the same response as above.)

Reviewer 3 Report

The manuscript “A Study on the Establishment of Physical Activity Environ-2 ment for People with Disabilities in South Korea” is an interesting read. Below are some edits for the authors to consider.

  1. The sentence on Page 2, line 51-52 is not very clear. “Moreover, PA of people with disabilities can prevent diseases and help to save the national health insurance fi-52 nance [9].”

It should be something like, “Moreover, PA participation among people with disabilities can prevent diseases and help to save the national health insurance fi-52 nance [9].”

  1. The following statement contains contradictory elements. On page 2, line 56, the sentence, “Most of the PA programs for people with disabilities in South Korea are conducted in public PA-related facilities..” 

Later on line 60-62, “Many people with disabilities participate in PA near or at home rather than at public PA-related facilities or social sports clubs for persons with disabilities..”

  1. On page 3, line 102, “The participants of the study were 32 experts with more than 5 years of experience on the PA for people with disabilities…” What does “on the PA” means? Can the authors illustrate what the experts study? Is it PA habit, intensity, or more like psychological behaviors?

  2. The sentence on Page 15, line 359-361 is grammatically incorrect. “According to the Disability Rights in the United Kingdom, inclusive the PA programs should be provided for persons with and without disabilities to participate in PA.”

It should be, “ “According to the Disability Rights in the United Kingdom, the inclusive PA programs..”

Author Response

(The authors gave the same response as above.)
